# Impact of surgical site infection (SSI) following gynaecological cancer surgery in the UK: a trainee-led multicentre audit and service evaluation

Rachel L O'Donnell,[1,2] Georgios Angelopoulos,[1,3] James P Beirne,[1,4] Ioannis Biliatis,[1,5] Helen Bolton,[1,6] Melissa Bradbury,[1,2] Elaine Craig,[1,4] Ketan Gajjar,[1,6] Michelle L Mackintosh,[1,7] Wendy MacNab,[1,8] Thumuluru Kavitha Madhuri,[1,9] Mark McComiskey,[1,7] Eva Myriokefalitaki,[1,10] Claire L Newton,[1,11] Nithya Ratnavelu,[1,2] Sian E Taylor,[1,12] Amudha Thangavelu,[1,13] Sarah A Rhodes,[14] Emma J Crosbie,[1,15] Richard J Edmondson,[1,15] Yee-Loi Louise Wan[15]

For numbered affiliations see end of article.

**Correspondence to**
Dr Yee-Loi Louise Wan;
louise.wan@manchester.ac.uk

## ABSTRACT

**Objectives** Surgical site infection (SSI) complicates 5% of all surgical procedures in the UK and is a major cause of postoperative morbidity and a substantial drain on healthcare resources. Little is known about the incidence of SSI and its consequences in women undergoing surgery for gynaecological cancer. Our aim was to perform the first national audit of SSI following gynaecological cancer surgery through the establishment of a UK-wide trainee-led research network.

**Design and setting** In a prospective audit, we collected data from all women undergoing laparotomy for suspected gynaecological cancer at 12 specialist oncology centres in the UK during an 8-week period in 2015. Clinicopathological data were collected, and wound complications and their sequelae were recorded during the 30 days following surgery.

**Results** In total, 339 women underwent laparotomy for suspected gynaecological cancer during the study period. A clinical diagnosis of SSI was made in 54 (16%) women. 33% (18/54) of women with SSI had prolonged hospital stays, and 11/37 (29%) had their adjuvant treatment delayed or cancelled. Multivariate analysis found body mass index (BMI) was the strongest risk factor for SSI (OR 1.08[95% CI 1.03 to 1.14] per 1 kg/m$^2$ increase in BMI [p=0.001]). Wound drains (OR 2.92[95% CI 1.41 to 6.04], p=0.004) and staple closure (OR 3.13[95% CI 1.50 to 6.56], p=0.002) were also associated with increased risk of SSI.

**Conclusions** SSI is common in women undergoing surgery for gynaecological cancer leading to delays in discharge and adjuvant treatment. Resultant delays in adjuvant treatment may impact cancer-specific survival rates. Modifiable factors, such as choice of wound closure material, offer opportunities for reducing SSI and reducing morbidity in these women. There is a clear need for new trials in SSI prevention

## Strengths and limitations of this study

► This study documents real-world clinical practice in the prevention and management of surgical site infection (SSI) in women undergoing open gynaecological cancer surgery in the UK.
► Data were collected prospectively and consecutively without exclusions according to comorbidities or treatment from all women undergoing gynaecological cancer surgery at 12 UK centres, encompassing the various national care settings (except Wales).
► Data relating to direct and indirect healthcare costs of SSIs were not available.
► Further longitudinal data examining the cancer-specific outcomes of this cohort are not currently available.

in this patient group; our trainee-led initiative provides a platform for their successful completion.

## INTRODUCTION

Gynaecological cancers collectively account for 11% of all cancers affecting women in the UK. Surgery remains the mainstay of treatment, and despite an increase in the use of minimally invasive techniques, a large proportion of patients still require laparotomy. For many women, their treatment will involve a hysterectomy with an associated high risk of surgical site infection (SSI) due to the inherently contaminated nature of procedures involving entry into the genital tract.

Although most women undergoing laparotomy can expect their wound to heal without any problem, there is a lack of high-quality data in gynaecological oncology to

indicate the incidence and consequences of SSI. A study by Iyer et al,[1] which aimed to identify predictors of complications in women undergoing surgery for gynaecological malignancy, describes grade II-IV infections in 27.8% of all gynaecological surgery cases; accounting for 31% of all complications. Although this study identified predictors of complications, the consequences of such complications were not assessed. The need to identify preventable complications coupled with effective interventions remains. The importance of SSI prevention, across all surgical specialities, has also been acknowledged by the UK government with the publication of various national guidelines outlining care bundles with recommended interventions during the preoperative, intraoperative and postoperative phases of the patient journey.[2 3] This has led to a downward trend in the incidence of SSIs in most types of surgery.[4] However, worryingly, there has been a year-on-year rise in SSI related to abdominal hysterectomies since 2014.[4]

Further research into the direct and indirect effects of SSI on the gynaecological oncology population and the additional interventions that could reduce SSIs is, therefore, urgently needed. An innovative UK-wide multicentre collaborative of trainees and consultants with an interest in clinical research in gynaecological oncology was established in 2015 to develop and implement evidence-based interventions to improve surgical care for women undergoing gynaecological cancer surgery.

The primary objective of this study was to define the incidence of SSI in the UK gynaecological oncology population undergoing laparotomy and record standard practices. The secondary objectives were to identify risk factors associated with infection and assess the potential impact on length of stay and delay to adjuvant cancer treatment. This study also aimed to provide baseline data to develop a modified enhanced recovery programme for women undergoing gynaecological cancer surgery.

## METHODS

This prospective multicentre audit of all consecutive women undergoing laparotomy for suspected or confirmed gynaecological malignancy was undertaken across 12 of the 44 UK tertiary centres during an 8-week period September–November 2015. Daily review of surgical cases was undertaken in each institute and cross referenced with theatre IT systems to ensure all subjects were included. Advice was sought from local ethics committees who confirmed that this study was exempt from ethical review as the research was limited to the secondary use of fully anonymised information previously collected in the course of normal care without the intention to use it for research at the time of collection.

Local audit leads were recruited via the British Gynaecological Cancer Society Trainees group and the Tomorrows Leaders in Surgical Research workshops. These local leads coordinated the prospective collection and coding of anonymised data using a standardised data collection tool developed in line with National Institute for Health and Care Excellence guidance.[2] Additional information including patient demographics, comorbidities, surgicopathological and wound complication data were also contemporaneously collected. All patients were categorised using the derived patient status risk index described by the American Society of Anesthesiologists (ASA).[5] Surgical complexity was classified using a modification of the surgical scoring system (online supplementary table A) developed by the Mayo Clinic[6] for ovarian cancer. Each individual procedure was given a score, and the total score for the surgery was the sum of the individual scores. Surgeries were categorised into three groups of increasing complexity based on the overall score (low: score ≤3; intermediate: score 4–8; radical: ≥8). In addition to the procedure, the duration of surgery and estimated blood loss were also recorded.

The Centres for Disease Control and Prevention (CDC) definitions for superficial incisional, deep incisional and organ/space SSI were used to standardise diagnosis of SSI across all centres (online supplementary table B).[7] A modified definition was applied for patient-reported SSI, as per the Public Health England Surgical Site Infection Surveillance programme protocol.[8] The impact of SSI on length of stay, return to theatre, use of negative pressure wound therapy, admission to intensive care and time to adjuvant chemotherapy or radiotherapy was studied.

To ensure completeness of data and to capture events following discharge, all recorded interactions (ie, telephone consultations/clinic visits/hospital records) with patients in the 30 days postoperatively were analysed. Data from patient reported infection following discharge as well as observed infection during admission were included in the analysis.

### Statistical analysis

Descriptive analysis of clinicopathological data were undertaken to characterise the cohort and the predictive value of each risk factor for SSI was separately assessed using univariable analysis by logistical regression or $\chi^2$ as appropriate. The variables to include in the logistic regression were based on the univariate analysis and known risk factors for SSI. To take into account variations in the length of time to last recorded contact, the incidence density was calculated using the number of cases in the numerator and the total number of days of patient follow-up (from inpatient surveillance) in the denominator giving the number of SSIs per 10 patient days of follow-up.

### Patient involvement

No patients were directly involved in this research. However, the views of focus groups were sought regarding the most appropriate sequelae of SSI to measure. Patients are actively involved in the design of potential interventions to prevent SSIs and any future trials from our trainee-led initiative.

## RESULTS

### Data collection

During the prospective study, undertaken between 1 September 2015 and 1 November 2015, with follow-up until 1 December 2015, 12 UK tertiary centres submitted data for a total of 339 patients. Data were complete in 98.0% of fields. Median time until last patient encounter was 22 days (IQR 14–41 days). The final patient contact in the postoperative follow-up period for most patients was at their outpatient review 4–6 weeks after surgery.

### Participants

Baseline patient characteristics are shown in table 1. The median patient age was 62 years (IQR 53–71) with a median body mass index (BMI) of 28.0 kg/m² (IQR 24.0–32.8). Preoperative performance status varied with 62 (18.3%) patients classified as ASA 1; 166 (49.0%) as ASA 2; and 87 (25.7%) as ASA 3. 13% of patients were hypoalbuminaemic (albumin <35 g/L).

The majority of surgeries were undertaken as primary treatments (244, 72%) with 69 (20.4%) patients undergoing interval surgery after neoadjuvant chemotherapy for ovarian cancer. Twenty-two (6.5%) patients underwent surgery for recurrent disease with palliative intent (from all primary sites). Of the 339 study patients, 288 (85.0%) had a diagnosis of cancer confirmed, the majority of whom, 200 (59%), had a diagnosis of ovarian cancer (table 1).

### Infection prophylaxis

All patients were screened for methicillin-resistant Staphylococcus aureus (MRSA) preoperatively, and colonisation with MRSA was confirmed in two patients who subsequently underwent preoperative eradication. Infection prophylaxis policy varied between centres with variations in surgical skin preparation and prophylactic antibiotics. A total of 318 out of 339 (93.8%) patients received intravenous antibiotics at induction of anaesthesia, of which 292/318 (91.8%) was in line with local microbiology policy. Of those who received prophylactic intraoperative antibiotics, 23 (7.2%) received triple therapy, 85 (26.7%) received combined double therapy and 210 (66.0%) received single agent prophylaxis. The most frequently used antibiotic was co-amoxiclav, used alone or in combination in 236 (74.2%) patients. Three hundred and six (90.3%) surgeries were classified as clean contaminated or contaminated.

### Surgery

The majority of laparotomy incisions were midline and extended above the level of the umbilicus to the pubis. Median operating time was 150 min (IQR 110–191 min). The median surgical complexity score for the entire cohort was 2 (range 1–17, IQR 2–3) (table 1). Seventy-four (21.8%) patients had a radicality score of ≥4, which included 50 (15.6%) patients who underwent bowel resection (table 1).

**Table 1** Patient demographics

| Characteristic | Cases (%) – unless otherwise stated |
|---|---|
| Age | n=339 |
| Age (years), median (IQR) | 62 (53–71) |
| BMI | n=323 |
| BMI (kg/m²), median (IQR) | 28.0 (24.0-32.8) |
| ASA | n=315 |
| 1 | 62 (18.3) |
| 2 | 166 (49.0) |
| 3 | 87 (25.7) |
| Comorbidity | n=334 |
| Diabetes | 34 (10.0) |
| Oral hypoglycaemic/insulin use | 21 (6.2)/13 (3.8) |
| Thyroid dysfunction | 43 (12.7) |
| Immune defect | 2 (0.6) |
| Preoperative anaemia | 34 (10.0) |
| Coagulopathy | 9 (2.7) |
| Chronic kidney disease | 15 (4.4) |
| Chronic lung disease | 37 (10.9) |
| Current smoker | 36 (10.6) |
| Other non-active malignancy | 16 (4.7) |
| Diagnosis | n=337 |
| Ovarian/primary peritoneal cancer | 200 (59.3) |
| Endometrial cancer | 61 (18.1) |
| Cervical cancer | 9 (2.7) |
| Benign | 51 (15.1) |
| Other non-gynaecological cancer | 16 (4.7) |
| FIGO stage | n=325 |
| 1 | 83 (25.5) |
| 2 | 23 (7.1) |
| 3 | 129 (39.7) |
| 4 | 24 (7.4) |
| Not applicable | 66 (20.3) |
| Surgical radicality | n=338 |
| Standard (≤3) | 265 (78.4) |
| Intermediate (4–8) | 61 (18.0) |
| Ultraradical (≥8) | 13 (3.8) |
| Procedure | n=338 |
| Hysterectomy | 272 (80.2) |
| Oophorectomy/salpingectomy | 305 (90.0) |
| Omentectomy | 250 (73.7) |
| Pelvic lymph node dissection | 67 (19.8) |
| Para-aortic lymph node dissection | 35 (10.3) |
| Small bowel resection | 19 (5.6) |
| Large bowel resection | 36 (10.6) |
| Peritoneal stripping | 55 (16.2) |

Continued

**Table 1** Continued

| Characteristic | Cases (%) – unless otherwise stated |
|---|---|
| Panniculectomy | 3 (0.9) |
| Colostomy | 15 (4.4) |
| Urostomy | 2 (0.6) |

ASA, American Society of Anesthesiologists, BMI, body mass index; FIGO, International Federation of Gynecology and Obstetrics.

### Postoperative wound management

Use of wound drains and skin closure techniques was highly variable and the incidence of drain use, skin closure and dressing used are shown in table 2. Dressing choice was largely dependent on National Health Service (NHS) site, with little variation within each institution.

### Surgical site infections

Of the 339 patients included in the study, 54 (15.9%) were diagnosed with an SSI by CDC criteria and 30 (8.8%) had wound dehiscence. Twenty (5.9%) patients had dehiscence in the presence of infection. To take into account the variation in the length of follow-up, incidence density was calculated; 0.5 cases of infection per 10 days of patient follow-up was seen. Severity of infection was variable with the incidence of various markers of severity shown in table 3. Infection was most prevalent in patients undergoing laparotomy for endometrial cancer, with an incidence of 21.3% in this subgroup. However, there was no significant difference in incidence of SSI by site of tumour origin (p=0.7358).

Causative micro-organisms were isolated from wound culture in only 32/54 cases (59.3%). The most prevalent culture result showing mixed growth predominantly of *Escherichia coli* and/or other bowel commensals. *Staphylococcus aureus* was a reported cause of SSI in four cases in this series.

**Table 2** Postoperative wound management

| Technique | | Usage n (%) | | Missing data (n) |
|---|---|---|---|---|
| Intra-abdominal drain | | 59 | 17.4 | 7 |
| Subcutaneous tissue closure | | 68 | 20.1 | 7 |
| Local anaesthetic wound catheter | | 16 | 4.7 | 7 |
| Skin closure | Staples | 174 | 51.3 | 0 |
| | Subcuticular monocryl | 164 | 48.4 | |
| | Other | 1 | 0.3 | |
| Wound dressing | Mepore | 145 | 42.8 | 2 |
| | Tegaderm | 68 | 20.1 | |
| | Opsite | 59 | 17.4 | |
| | Other | 63 | 18.6 | |

The incidence of infection per cancer centre was variable ranging from 6.3% to 41.2%. A funnel plot was constructed for each centre using the SSI incidence per centre (figure 1).

Further review of the centre that appears to be an outlier (figure 1, red triangle) demonstrates no evidence that the threshold for diagnosis of SSI was any lower at this centre. There was no excess of preoperative comorbidities in this subgroup of patients. However, there was a statistically significant difference in the radicality score of those with infection from this centre in comparison with the other 11 centres, with a mean score of 4.29 versus 2.38, p=0.0194.

### Predictors of SSI

Univariate analysis suggested that higher BMI, ASA, diabetes requiring insulin, use of a drain, choice of antiseptic for skin preparation and skin closure were associated with significant increase in SSI risk (p<0.05) (table 4). After accounting for other variables, higher BMI was associated with having a greater risk of infection, with the odds of infection increasing by on average 8% (95% CI 3% to 14%) for each extra unit of BMI. Insertion of a wound drain and wound closure with staples rather than suture were all associated with having an increased risk of infection (table 4). There was no evidence to suggest that ASA or using betadine rather than chlorhexidine skin preparation were associated with an increased infection rate, after taking into account the other variables.

### Sequelae

Median length of postoperative hospital stay was 5 days (range 1–71, IQR 4–8). In those that developed infection, there was clinical delay in discharge in 18/54 (33.3%). In the 37 patients with SSI who required adjuvant chemotherapy or radiotherapy, 11 (29.7%) experienced delay in commencing therapy directly or indirectly as a result of infection.

Approximately half of all infections were diagnosed following discharge and 5 (1.5%) patients necessitated readmission to hospital following diagnosis of infection. Thirty (55.6%) patients had evidence of skin dehiscence, 14 of which required wound packing. As a result of wound complications, four patients returned to theatre, one patient was admitted to level 2/3 care and two patients required negative pressure wound therapy.

### DISCUSSION

Modelled on the highly successful trainee-led research collaboratives in other surgical specialties, the surgical gynaecological oncology research network was formed in 2015 to address the paucity of clinical research in gynaecological cancer surgery. Our study provides a snapshot of current practice of the preoperative, intraoperative and postoperative prevention and management of SSI in women undergoing open gynaecological oncology surgery across the UK.

**Table 3** Markers of surgical site infection (SSI) severity

| Infection type and severity, n=54 | | | Cases (%), unless otherwise stated |
|---|---|---|---|
| Infection | Cultured organisms | Positive culture | 32 (59.3) |
| Time of infection | Postoperative admission | 24 (44.4) | |
| | Community | 25 (46.3) | |
| | Local hospital readmission | 4 (7.4) | |
| | Central hospital readmission | 1 (1.9) | |
| Antibiotic route | Intravenous | 24 (44.4) | |
| | Oral | 25 (46.3) | |
| | None | 6 (11.1) | |
| Length of course | Days, median (range) | 7 (5–14) | |
| Severity | Fever | Temperature >38°C | 11 (20.4) |
| | WCC | ×10/L, median (IQR) | 10 (7–14) |
| | Cellulitis | Clinical diagnosis | 24 (44.4) |
| | Skin dehiscence | Clinical diagnosis | 30 (55.6) |
| | | Treated with packing | 14 (25.9) |
| | | Managed without packing | 16 (29.6) |
| | Sheath dehiscence | | 1 (1.9) |
| | Intensive care unit/High dependency unit admission | | 1 (1.9) |
| | Negative pressure wound therapy | | 2 (3.7) |
| | Return to theatre | | 4 (7.4) |

WCC, white cell count.

Our study demonstrates that SSI is highly prevalent within the gynaecological oncology population who undergo laparotomy with an incidence of 15.9%; 10 times higher than the incidence reported in the NHS Surgical Site Infection Surveillance Programme for women undergoing abdominal hysterectomy in 2015/2016.[9] The high

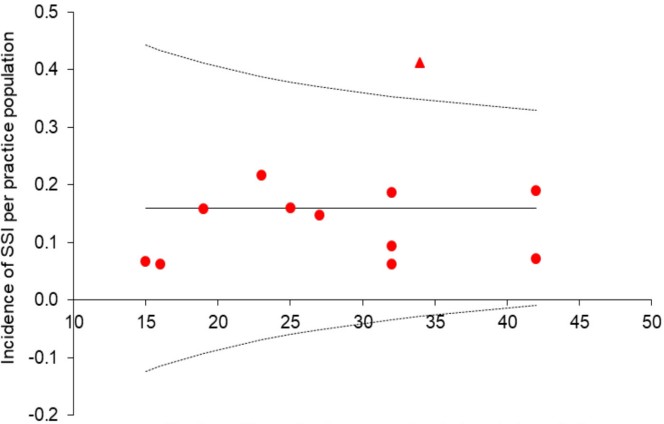

**Figure 1** Funnel plot of surgical site infection (SSI) by UK cancer centre: funnel plot of the number of reported SSIs for each gynaecological cancer unit versus effective sample size for each unit. The funnels correspond to 95% confidence limits (dashed lines). The outlier unit is denoted by a triangle, while units with SSI incidence within the 95% confidence limits are marked with a circle.

burden of complications is likely to reflect a combination of patient, surgical and disease-specific factors. Unsurprisingly, infection is prevalent in patients with a high BMI, which perhaps explains the greater incidence observed in those within the endometrial cancer subgroup and in those with diabetes, given shared aetiological factors. Cancer and its treatments can result in immune deficiency, increasing the risk of SSI in oncology patients. The contaminated nature and complexity of the procedures required in gynaecological oncology surgery further increase this risk.[10]

In a small proportion of patients, SSI is severe and/or life threatening, necessitating return to theatre or advanced wound care and prolonged hospital stays. Our study provides evidence that SSI impacts on time to delivery of adjuvant cancer treatments in a significant proportion of patients. This has the potential to impact on response to treatment and overall cancer specific survival in these patients.[10–12] Additional data, however, are needed to evaluate the impact of these delays on progression and mortality.

In this study, infection occurred in patients even where compliance with local infection prophylaxis standards were high. Analysis of the micro-organisms cultured from patients affected showed that *E. coli* and *Enterobacter* species predominated. This is in keeping with other studies of wound infection post hysterectomy[13–15]

**Table 4** Predictors of SSI

| Type of risk factor | Variable | OR (95% CI) | P value |
|---|---|---|---|
| **Univariate logistic regression** | | | |
| Non-modifiable prior to surgery | BMI (per kg/m$^2$) | 1.09 (1.04 to 1.13) | <0.001 |
| | Age (per year) | 1.01 (0.99 to 1.04) | 0.236 |
| | ASA three or more (vs less than three or missing) | 1.73 (0.93 to 3.22) | 0.083 |
| | Albumin (per g/L) | 0.98 (0.93 to 1.02) | 0.309 |
| | FIGO stage (per unit) | 1.05 (0.76 to 1.45) | 0.773 |
| | Diabetic requiring insulin (vs not requiring insulin for glycaemic control) | 5.00 (1.61 to 15.5) | 0.005 |
| | Neoadjuvant chemotherapy (vs none) | 0.50 (0.22 to 1.17) | 0.110 |
| | Cancer type | | |
| | Endometrium (vs ovarian) | 1.48 (0.72 to 3.04) | 0.291 |
| | Benign (vs ovarian) | 0.68 (0.08 to 5.64) | 0.722 |
| | Cervical (vs ovarian) | 0.87 (0.36 to 2.10) | 0.752 |
| | Other (vs ovarian) | 0.73 (0.16 to 3.34) | 0.682 |
| Non-modifiable surgical risk factor | Surgical radicality score (per unit) | 1.02 (0.89 to 1.19) | 0.718 |
| | Clean contaminated (vs clean) | 2.26 (0.67 to 7.66) | 0.189 |
| | Incision site | | |
| | Above umbilicus (vs transverse) | 1.64 (0.36 to 7.39) | 0.519 |
| | Below umbilicus (vs transverse) | 2.15 (0.46 to 10.19) | 0.333 |
| Modifiable at time of surgery | Chlorhexadine skin preparation (vs betadine) | 0.54 (0.30 to 0.97) | 0.039 |
| | Wound drain (vs none) | 3.23 (1.68 to 6.19) | <0.001 |
| | Subcutaneous fat stitch (vs none) | 1.50 (0.76 to 2.95) | 0.245 |
| | Staples (vs subcuticular suture) | 3.58 (1.84 to 6.97) | <0.001 |
| | Prophylactic postoperative antibiotics (vs none/missing) | 1.69 (0.86 to 3.31) | 0.126 |
| **Multivariate logistic regression** | | | |
| Non-modifiable prior to surgery | BMI (per kg/m$^2$) | 1.08 (1.03 to 1.14) | 0.001 |
| | ASA (per unit) | 1.01 (0.48 to 2.11) | 0.989 |
| | Diabetic requiring insulin (vs not requiring insulin for glycaemic control) | 3.34 (0.90 to 12.39) | 0.071 |
| Modifiable at time of surgery | Chlorhexadine (vs betadine) | 0.58 (0.30 to 1.134) | 0.109 |
| | Drain (vs none) | 2.92 (1.41 to 6.04) | 0.004 |
| | Staples (vs subcuticular suture) | 3.13 (1.50 to 6.56) | 0.002 |

ASA, American Society of Anesthesiologists; BMI, body mass index; SSI, surgical site infection.

and the vaginal flora of our, predominately, postmenopausal cohort.[16] Previous studies have determined that up to 15% of cultured organisms may be resistant to broad spectrum combinations of antibiotics.[15] In our study, sensitivities of the microorganisms cultured from affected patients were not readily available. Further studies would be required to determine whether resistance to current antibiotic regimens could explain the high prevalence of SSI in this population.

The use of drains and staples for wound closure was positively associated with SSI in our population. These devices are commonly used in patients thought to be at high risk of SSI, and therefore, the associations seen in this study may not be causative. Surgical drains are often used to drain body fluids away from a dead space to promote wound healing and prevent potential SSI. Although there

is conflicting data as to whether the drains are associated with an increase in SSI, no studies have shown a reduction in SSI following the use of surgical drains.[17–20] Judicious use of drains and timely removal is therefore indicated. Staples are an efficient way to close a linear incision and may be preferred by some surgeons after particularly long or technically difficult cases. Various randomised control studies and subsequent meta-analyses fail to show that the use of sutures over staples resulted in any difference in SSI in gynaecological surgery.[21–25]

At a time when many NHS trusts are keen to invest in enhanced recovery programmes to encourage a speedy and proactive approach to recovery for patients, this audit demonstrates a need for effective management of wounds to minimise the incidence of wound infection and provide effective management rapidly when it does

occur. These data support the development of a stratified care bundle to reduce delay to adjuvant treatment associated with SSI focusing on women with high BMI and poorly controlled diabetes.

## Strengths and limitations

To the best of our knowledge, this study is the first to comprehensively review current practice surrounding the prevention and management of SSI in open gynaecological cancer surgery. The United Kingdom Gynaecological Oncology Surgical Outcomes and Complications (UKGOSOC) study found a slightly lower wound complication rate of 9.6% across all types of gynaecological cancer surgery in ten UK gynaecological cancer centres.[1] However, this study included patients undergoing both laparoscopic and open surgery. Laparoscopic surgery has previously been shown to have lower incidence of postoperative complications such as wound infection.[26–28] We have, therefore, chosen to examine care practices in women undergoing open surgery specifically.

The strength of our review of practice is that the data were prospectively collected using standardised data collection forms at 12 participating centres from around the UK. As this was a review of practice, data from all consecutive surgeries were included as patient recruitment was not required. We included all classes of SSI (including superficial incisional infections) and their sequelae; thus, not presupposing, that particular classes of SSI would have an impact on patient recovery and subsequent cancer treatment. Despite our use of a standardised proforma and the use of a preagreed definition of SSI, there remained a potential for variation in its interpretation from site to site. This may have contributed to the outlier unit with the disproportionately high rate of SSI. However, this type of variation is likely to also reflect that seen in day-to-day practice. Twelve of the 44 units that offer tertiary level gynaecological cancer surgery participated in this review. No units from the South West or Wales participated. The inclusion of these units together with a longer data collection window may have led to a larger sample, allowing the identification of further potential risk factors for SSI and highlighting additional practices that may reduce wound infection rates. The identification of risk factors and good practice points will enable us to develop a tool to personalise patient care plans with the aim of reducing SSI.

## CONCLUSIONS

This trainee-led collaborative initiative demonstrates the need to determine whether addressing modifiable risk factors can prevent SSI in gynaecological cancer surgery. Ultimately, a reduction in delayed discharge and subsequent delays in the commencement of adjuvant treatment could result in improved patient outcomes.

**Author affiliations**

$^1$SGRN, Surgical Gynaecological Oncology Research Network, UK
$^2$Northern Gynaecological Oncology Centre, Gateshead Foundation NHS Trust, Newcastle, UK
$^3$Gynaecological Oncology, James Cook University Hospital, Middlesborough, UK
$^4$Northern Ireland Centre for Gynaecological Oncology, Belfast City Hospital, Belfast, UK
$^5$Gynaecological Oncology, Royal Marsden NHS Foundation Trust, London, UK
$^6$Gynaecological Oncology, Addenbrooke's Hospital, Cambridge, Cambridgeshire, UK
$^7$St Mary's Hospital, Manchester University Hospitals NHS Foundation Trust, Manchester, UK
$^8$Gynaecological Oncology, Glasgow Royal Infirmary, Glasgow, UK
$^9$Gynaecological Oncology, Royal Surrey County Hospital, Guildford, UK
$^{10}$Gynaecological Oncology, University Hospital Leicester, Leicester, UK
$^{11}$Gynaecological Oncology, St Bartholomew's Hospital, London, UK
$^{12}$Gynaecological Oncology, Liverpool Women's Hospital, Liverpool, UK
$^{13}$Gynaecological Oncology, Nottingham University Hospital, Nottingham, UK
$^{14}$Centre for Biostatistics, Division of Population Health, Health Services Research and Primary Care, University of Manchester, Manchester, UK
$^{15}$Division of Cancer Sciences, Faculty of Biology, Medicine and Health, University of Manchester, Manchester, UK

**Contributors** RLO, YLW, MB, TKM, AT, MLM, MM, EJC and RJE conceived and designed the study. Data were collected by RLO, NR, GA, MB, IB, HB, JPB, EC, KG, WM, TKM, MLM, MM, EM, CLN, NR, SET and AT and analysed by RLO, MM, TKM, SAR and YLW. The article was drafted by RLO and YLW and revised and approved by all authors. All authors agree to be accountable for all aspects relating to the accuracy and integrity of the work.

**Funding** YLW was funded by the Wellcome Trust as part of a Wellcome Trust Clinical Research Fellowship (101714/Z/13/Z). EJC is a National Institute for Health Research (NIHR) Clinician Scientist (NIHR-CS-012-009) and is supported by the NIHR Manchester Biomedical Research Centre. Patient involvement was funded by a NIHR North West Research Design Service Patient and Public Involvement Bursary. Neither funder was involved in the planning or execution of the study or the drafting of the manuscript.

**Competing interests** None declared.

**Patient consent for publication** Not required.

**Ethics approval** This study was exempt from ethical review as the research was limited to the secondary use of fully anonymised information previously collected in the course of normal care without the intention to use it for research at the time of collection

**Provenance and peer review** Not commissioned; externally peer reviewed.

**Data sharing statement** Original data are available from the corresponding author on request.

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
