## [Reviewer comments · BMJ Open]

This paper was submitted to a another journal from BMJ but declined for publication following peer review. The authors addressed the reviewers' comments and submitted the revised paper to BMJ Open. The paper was subsequently accepted for publication at BMJ Open.

(This paper received three reviews from its previous journal but only two reviewers agreed to published their review.)

ARTICLE DETAILS

TITLE (PROVISIONAL)	IMPACT OF SURGICAL SITE INFECTION (SSI) FOLLOWING GYNAECOLOGICAL CANCER SURGERY IN THE UNITED KINGDOM: A TRAINEE-LED MULTICENTRE AUDIT AND SERVICE EVALUATION
AUTHORS	O'Donnell, Rachel; Angelopoulos, Georgios; Beirne, James; Biliatis, Ioannis; Bolton, Helen; Bradbury, Melissa; Craig, Elaine; Gajjar, Ketan; Mackintosh, Michelle; MacNab, Wendy; Madhuri, Thumuluru Kavitha; McComiskey, Mark; Myriokefalitaki, Eva; Newton, Claire; Ratnavelu, Nithya; Taylor, Sian; Thangavelu, Amudha; Rhodes, Sarah; Crosbie, Emma; Edmondson, Richard; Wan, Yee-Loi

VERSION 1 – REVIEW

REVIEWER	Keita Morikane Yamagata University, Japan
REVIEW RETURNED	09-Aug-2018

GENERAL COMMENTS	Epidemiology, risk factors and practices associated with SSI following surgery of gynecological malignancy is poorly investigated. This paper adds significant insight to this issue, and authors should be well applauded. One concern is that the paper is a bit too lengthy with too many data presented in one paper. Specifically, Table 3 and associated discussion might be omitted.
--

REVIEWER	Miquel PUJOL Bellvitge University Hospital Infectious Disease Service Barcelona, Spain
REVIEW RETURNED	24-Aug-2018

GENERAL COMMENTS	Thanks you for allowing me to review. Very interesting and well-written. This is an interesting article, which demonstrates the impact of surgical infection not only on the prolongation of the stay and readmissions but also on the delay or rejection for adjuvant treatment of the neoplastic disease. This is little known and it is important to emphasize. I believe that it is important to publish these results . Abstract: well summarized, adequately presents the data. The conclusions are consistent
--

	Introduction: No comments. The main objective and secondary objectives are well expressed Methods: Line 128: Supplementary Table A and reference 6. There is a mistake in the table A. According to CDC criteria, third row of the table should define organ/space infections. Also reference 6 is not adequate: I would like to use the following one: https://www.cdc.gov/nhsn/pdfs/psscmanual/9psscscurrent.pdf Results: The infection rate is very high. It is probably because superficial incisional infection was included. Although the methodology of the study followed the CDC definitions of surgical site infections, the classification of SSI in the study is unclear. For example cases with organ / space infections did not appear. Maybe they were not diagnosed in the cohort, but still it is necessary to explain these results. On the other hand, it is important to differentiate superficial incisional infection from deep incisional infection given that the former has no clinical impact The risk factors for infection are consistent and in line with other publications except staples. It is striking that the majority of infections are caused by E. coli, a pathogen from the digestive tract and that is not usually observed in the gynecological tract. On the other hand infection by S.aureus is much reduced. Do we have some information about the type of systemic antibiotic prophylaxis that was administered to patients? The coverage of antibiotic prophylaxis included E.coli? Discussion It should be necessary to discuss in greater depth the risk factors of infection, especially drainages and staples. How infection rates can be reduced and why contamination of the surgical wound by E.coli occurs in most cases. It is during surgery or is related to postoperative care and drainage. What is the role of drainages in the infection? Suture with staples is not usually a well-established risk factor. What explanation do the authors have?
--	--

VERSION 1 – AUTHOR RESPONSE

In response to reviewer 1, we have shortened the length of the results section by removing duplications of data displayed in Tables 2 & 3 from the text in the body of the manuscript and removed the discussion of the health care related costs associated with SSI. We feel however, that Table 3, which documents the sequelae of the SSI, is fundamental to the paper itself and demonstrates the range of complications and interventions that patients experience.

In response to reviewer 2, we have updated the reference used for the CDC criteria of SSI accordingly and corrected the omission that they kindly noted in Supplementary Table A. We welcomed the suggestion that we may have wished to consider the subclasses of SSI separately. In this study, we chose to consider SSI as a single outcome. We preferred not to assume that the subclassification of the type of SSI correlated directly with its clinical impact. We instead looked at markers of SSI severity, as surrogate markers for clinical impact. We feel that such markers may be more useful in understanding impact as they are factors which might make a clinician consider additional treatment (e.g. pyrexia, dehiscence) and/or might delay ongoing treatment/adjunct treatment (e.g. return to theatre, ITU admission, VAC therapy).

Reviewer 2 also noted that the risk factors we describe for SSI are in line with other publications except staples. We agree, it is striking that this association holds even in our multivariate regression. However, as this was an observational study and not a randomised controlled trial, it is subject to potential confounders. Within the surgical community, there remains a proportion of surgeons who

continue to use staples, rightly or wrongly, in patients thought to be at high risk of SSI. This association may result from this practice.

Regarding the comment on the predominance of E-coli grown in the cultures of affected patients, we commonly culture bowel pathogens from the wounds and collections of affected patients. This has been documented in both our study and those of others. These pathogens are also commonly found within the vaginal flora of our patient population. As we have no record of the sensitivities of the positive cultures, we cannot comment as to whether the antimicrobial regime appropriately covered these pathogens. However, we do describe the antibiotic regimes used in our population in the 'Infection prophylaxis' subsection of the Results section. These provide broad spectrum cover likely to cover most gram-negative rods and anaerobes. Our data support the need for a stratified care bundle that goes above and beyond that currently used as standard of care. A specific focus on improving SSI rates in obese women and the judicious use of drains and staples in these high-risk women is advocated.

VERSION 2 – REVIEW

REVIEWER	Keita Morikane Yamagata University Hospital Japan
REVIEW RETURNED	07-Nov-2018
GENERAL COMMENTS	I appreciate authors' effort to shorten and simplify the manuscript as much as possible.
REVIEWER	Miquel PUJOL Bellvitge University Hospital
REVIEW RETURNED	07-Nov-2018
GENERAL COMMENTS	I agree with this reviewed version. Thank you very much for your effort